# Healthcare Burden of Rare Diseases: A Population-Based Study in Tuscany (Italy)

**DOI:** 10.3390/ijerph19137553

**Published:** 2022-06-21

**Authors:** Silvia Baldacci, Michele Santoro, Anna Pierini, Lorena Mezzasalma, Francesca Gorini, Alessio Coi

**Affiliations:** 1Unit of Epidemiology of Rare Diseases and Congenital Anomalies, Institute of Clinical Physiology, National Research Council, 56124 Pisa, Italy; michele.santoro@ifc.cnr.it (M.S.); apier@ifc.cnr.it (A.P.); lorena.mezzasalma@ifc.cnr.it (L.M.); fgorini@ifc.cnr.it (F.G.); alessio.coi@ifc.cnr.it (A.C.); 2Fondazione Toscana Gabriele Monasterio, 56124 Pisa, Italy

**Keywords:** rare diseases, disease registry, hospitalisation, length of stay, data linkage

## Abstract

Patients with rare diseases (RDs) need tailored, continuous, and multidisciplinary hospital care. This retrospective cohort study aimed to analyse the healthcare burden of RD patients using a multi-database approach, by linking the data of the Rare Diseases Registry of Tuscany with the regional hospital discharge database. The study population included 21,354 patients diagnosed with a RD between 1 January 2000 and 31 December 2017. The healthcare burden was evaluated for all the RDs during 2009–2018 period. The hospitalisation rate (per 1000) decreased over the years, ranging from 606.9 in 2009 (95% CI: 589.2–625.0) to 443.0 in 2018 (95% CI: 433.2–453.0). A decrease in the average length of stay (LOS) was observed in the earlier years, followed by an increase up to a steady trend (8.3 days in 2018). The patients with RDs of metabolism and the genitourinary system showed the highest hospitalisation rate (903.3 and 644.0 per 1000, respectively). The patients with rare immune system disorders and diseases of the skin and subcutaneous tissue showed the highest LOS (9.7 and 9.5 days, respectively). The methodological approach presented in this population-based study makes it possible to estimate the healthcare burden of RDs, which is crucial in the decision-making and planning aimed at improving patient care.

## 1. Introduction

Rare diseases (RDs) are a public health priority [1]. According to the Regulation of the European Commission (No. 141/2000), a RD is defined as a condition affecting no more than 5 per 10,000 people [2,3]. There are currently up to 8000 described RDs and, although individually rare, these disorders collectively affect 6–8% of the European population, corresponding to over 30 million people [4,5]. Due to their rarity and lack of clinical expertise, the early diagnosis of RDs is often difficult, resulting in delayed access to hospital care and pharmacological therapy [6].

RDs are severe, multi-systemic, often life-threatening or chronically debilitating, and capable of impairing physical and mental abilities and shortening life expectancy. Most of them have a genetic aetiology, often with an onset at birth or in childhood, making them leading causes of hospitalisations in the paediatric population [7,8,9,10].

RDs have a considerable impact on affected individuals, their families, health care providers, and the community [11]. Patients with RDs need tailored, continuous, highly specialised, and often multidisciplinary hospital care. RDs are therefore a major public health issue and their primary prevention, as well as ensuring access to safe diagnosis and quality care, should be a priority for public health policies worldwide [12,13].

However, the real burden of RDs is difficult to estimate, mainly due to the paucity of reliable population data. In addition, the lack of proper coding for many RDs in administrative healthcare data makes it difficult to identify and retrieve information on hospitalised cases [14,15].

Few studies have analysed the hospitalisation of RD patients at population level. Such studies are more often based on hospital-based cohorts or on cohorts extracted from hospital administrative data rather than based on population data [16,17,18,19,20,21].

The use of population-based registries is an effective tool to provide available data for epidemiological studies, and to support the healthcare management of RDs through the production of epidemiological indicators based on a specific geographical area [22,23]. These registries are often characterised by a high degree of case ascertainment, but they usually lack clinical data.

The aim of this population-based study was to characterise the healthcare burden of RD patients in terms of hospitalisation rate, length of stay (LOS), inpatient proportions, and main discharge diagnosis, using a multi-database approach to integrate data collected by a population-based RD registry, with hospital discharge records, routinely collected at local level.

## 2. Materials and Methods

### 2.1. Study Design and Population

This was a retrospective cohort study. The study population included all patients with a RD, residing in Tuscany, an Italian region of 3,701,343 inhabitants (source: Italian National Institute of Statistics as of 1 January 2018), diagnosed between 1 January 2000 and 31 December 2017.

### 2.2. Data Source

RD cases were extracted by the population–based Rare Diseases Registry of Tuscany (RDRT). RDRT monitors all RDs according to Italian Law (Decree of the President of the Council of Ministers, 01/2017), with a specific exemption from co-payment. Overall, 509 RDs, classified into 16 nosological groups, were included in the study (see Appendix A). The registry has been active since 2005 and is based on a regional network, allowing the detection of all cases diagnosed at any age by any of the regional health centres, as well as being one of the main contributors to the National Centre of Rare Diseases of the Italian National Institute of Health [24].

All RD cases, endowed with a unique regional anonymous identification number, were linked to regional mortality database; the Registry Office database was also used to identify the patients, in the cohort, who had died or moved from the study area during the study period. In this way, the exact number of prevalent cases was calculated for each year. Next, the prevalent cases were linked to the regional Hospital Discharge Database (HDD), exploiting all the discharges registered between 1 January 2009 and 31 December 2018 (Figure 1).

The HDD collects data related to the admissions of all the inpatients accessing care in the Italian National Health Service. In this study, inpatients were defined as RD patients with at least one hospital discharge during the study period. Admissions were defined as ‘ordinary’, when the inpatient stayed overnight in hospital (i.e., LOS ≥ 1) and as a ‘day-case’, when the inpatient was admitted to receive planned medical or paramedical services without staying overnight in hospital.

Both the diagnoses and the procedures were recorded according to the International Classification of Diseases, 9th Revision, Clinical Modification (ICD-9-CM).

Each hospital discharge record, according to inpatient’s age, gender, main diagnosis, secondary diagnoses, surgical or not surgical procedures performed, and status of the patients at discharged, is assigned to a specific Diagnosis-Related Group (DRG) [25]. In 1994, the Ministry of Health adopted the DRG for the inpatients Reimbursement of the Italian National Health Service (DM 15 April 1994). DRGs are classified into two groups: surgical and medical. In this study, we used this information to identify patients undergoing to a surgical procedure (DRG surgical) and those on whom a surgical treatment was not performed (DRG medical).

### 2.3. Study Outcomes and Data Analysis

Two lines of analysis were carried out: analyses performed on the whole cohort (all RDs investigated) and analyses performed by RD group.

For the whole cohort, ten-year (2009–2018) hospitalisation trends, referred to as hospitalisation rate (for overall, ordinary and day-case admissions), LOS, and inpatient proportions were calculated. The distribution of the main discharge diagnosis in 2018, identified by the principal discharge diagnosis ICD-9-CM codes, was also analysed. For these analyses, if a patient had more than one RD, she/he was counted only once.

More detailed indicators of hospitalisation were calculated to evaluate the healthcare impact for each RD group in the cohort of RD patients prevalent at 1 January 2018, who had a hospital discharge between 1 January and 31 December 2018. In particular, proportions of inpatients, total number of admissions by type (ordinary/day-case), hospitalisation rate (with 95% confidence interval, 95% CI), and average LOS were calculated. Additionally, the distribution by type of admission (i.e., ordinary, day-case), emergency admissions, and DRG type (surgical/medical) were calculated by each RD group. RD groups with ten or fewer RD cases (i.e., infectious and parasitic diseases and neonatal morbidities of perinatal origin groups) were not included in this analysis. Therefore, analyses by RD group were performed on 14 nosological groups. For these analyses, if a patient had more than one RD in different RD groups, she/he was counted in each RD group.

Hospitalisation rate (per 1000 cases) was calculated as the ratio between the total number of discharges occurring during the year and the total number of prevalent RD patients at 1 January of each year.

Average LOS was computed for ordinary admission only as the ratio between the total number of hospital days (date of discharge minus date of admission) by the number of discharges.

A Poisson regression model was also used to test time trends in hospitalisation. 

A *p*-value less than 0.05 was considered statistically significant when performing statistical analyses. 

Statistical analyses were conducted using Stata, version 16 (StataCorp LP, College Station, TX, USA).

## 3. Results

### 3.1. Description and Time Trend of Hospitalisation of the Cohort

The study involved 21,354 patients diagnosed with a RD between 1 January 2000 and 31 December 2017 (males: 9795, 45.9%; females: 11, 559, 54.1%). The mean age at diagnosis was 43.2 ± 25.4 (males: 41.5 ± 26.2; females: 44.6 ± 24.6).

In the study period (1 January 2009–31 December 2018), 93,605 hospital discharges were observed (50,914 ordinary admissions and 42,691 day-case admissions) with an average LOS of 8.1 days. 

The trend of the hospitalisation rate, for the whole cohort significantly decreased (*p* < 0.001) from 606.9 per 1000 cases in 2009 (95% CI: 589.2–625.0) to 443.0 per 1000 cases in 2018 (95% CI: 433.2–453.0) (Figure 2).

For the ordinary admissions, the hospitalisation rate slightly decreased, showing a steady trend from 2014 to the end of the study period. For the day-case admissions, the rate reached a maximum of 323.9 in 2011 (95% CI 312.5–335.7) and then decreased in the following seven years, down to 198.7 (95% CI 192.2–205.4) in 2018.

The inpatient proportions decreased from 34.5% (2009) to 24.7% (2018) (Figure 3). 

In particular, a significantly higher percentage of females among the inpatients was observed for the first three years of the study period (*p* < 0.05), while, subsequently, no significant differences were observed.

Concerning the average LOS, a decreasing trend was observed up to 2014 (7.8 days), followed by an increase up to a quite steady trend in the last years (8.3 days in 2018) (Figure 4). 

Concerning the main causes of hospitalisation, the time trend remained almost stable for all the ICD-9-CM groups, with two exceptions. In particular, the main discharge diagnoses due to endocrine, nutritional and metabolic diseases, and immunity disorders showed an increasing trend from 2009 (7.2%) to 2013 (11.7%), followed by a steady trend until 2016 and a decrease until the end of the study period, with percentages ranging between 11.3% and 13.8% (Figure 5a). By contrast, a decreasing trend was observed for the main discharge diagnoses due to diseases of the musculoskeletal system and connective tissues from 2015 (10.3%) to 2018 (5.9%) (Figure 5b).

Table 1 shows the frequency distribution of the main discharge diagnosis for all the inpatients in the last year of study (2018).

The most common main discharge diagnoses were of diseases of the nervous system and sense organs (12.8% of the total number of discharges) and to endocrine, nutritional and metabolic diseases, and immunity disorders (11.3%). In terms of the ordinary admissions, the most common main discharge diagnoses were of diseases of the respiratory and circulatory system (16.4% and 13.7%, respectively). For the day-case admissions, the most frequent main discharge diagnoses were of endocrine, nutritional and metabolic diseases, immunity disorders (20.8%), and diseases of the nervous system and sense organs (16.8%).

### 3.2. Hospitalisation by Rare-Disease Group

The number of RD patients at 1 January 2018 were 17,608, for a total of 7801 admissions between 1 January and 31 December 2018. 

The highest proportion of inpatients was reported for patients with RDs of the metabolism (42.9%) while, the lowest, was registered among individuals with RDs of the eye and adnexa (8.6%) (Table 2). 

No significant differences between males and females were observed in the inpatient proportions (data not shown) except for a higher percentage of females in the group of rare metabolic diseases (46.3% vs. 40.4% for males, *p* = 0.03) and in the group of diseases of the eye and adnexa (10.8% vs. 7.0% for males, *p* = 0.02).

In 9 out of the 14 analysed RD groups, there was a higher proportion of ordinary than of day-case admissions. The highest proportion of ordinary admissions was observed for the RD groups of respiratory diseases (82.1%) and digestive disorders (80.6%). The day-case admissions more frequently occurred in the patients with rare metabolic diseases (68.5%) (Table 2).

Among the ordinary admissions, the highest frequency of emergency admissions was observed in the patients with rare metabolic diseases (63.4%). For 8 out of the 14 groups of RDs, more than half of the ordinary admissions were emergency admissions. Conversely, the lowest proportions of emergency admissions were observed for the patients with rare neoplasms, endocrine diseases, and congenital anomalies (38.0%, 38.0%, and 40.1%, respectively).

The average LOS ranged from 5.5 to 9.7 (Table 2). In particular, the highest average LOS was observed for the inpatients with rare immune system disorders, RDs of the skin and subcutaneous tissue, peripheral and central neural disorders, and digestive disorders (9.7, 9.5, 9.4 and 9.3, respectively). By contrast, inpatients with rare disorders of the eye and adnexa showed the lowest average LOS.

All the RD groups were characterised by a markedly higher percentage of medical than surgical DRGs, except for the inpatients with RDs of the eye and adnexa, 52% of whom required surgical procedures (Table 2). 

The group of rare metabolic diseases had the highest overall hospitalisation rate (903.3 per 1000), while the group of rare disorders of the eye and adnexa had the lowest (107.7 per 1000) (Table 3). 

Among the ordinary admissions, the highest and lowest hospitalisation rates were observed in the group of respiratory diseases (411.6 per 1000) and in patients with disorders of the eyes and adnexa (74.1 per 1000), respectively. 

The group of metabolic diseases reported the highest day-case hospitalisation rate, while the lowest was detected in the group of rare disorders of the eyes and adnexa (619.0 per 1000 and 33.6 per 1000, respectively). 

## 4. Discussion

RDs are heterogeneous life-threatening conditions associated with multimorbidity and long-term disability. RDs require multidisciplinary, chronic, ongoing hospital care, as well as highly specialised and tailored care pathways, which exert an extremely high social and economic burden [26]. Because of their rarity, RDs represent significant challenges for affected patients, their families, and for clinicians attempting to define diagnoses and implement the best care [27,28]. Moreover, the lack of epidemiological and clinical data is a major issue in health-service planning for RDs [14].

The whole cohort of prevalent RD cases involved in this study, representing less than 1.0% of the Tuscany population, had a higher number of hospital discharges and a longer average LOS than the general population of Tuscany. In particular, the overall hospitalisation rates in the RD population were from four-to three-fold higher than those of the Tuscany population (154.0 and 134.8 in 2009 and 2018, respectively), and the average LOS was two days longer compared to that of the general population (6.6 days and 6.5 days in 2009 and 2018, respectively) [29]. The greater hospitalisation rate of the patients with RDs was expected and was the consequence of several factors, such as the chronicity of the diseases, disease-related complications, and frequent side effects related to therapy regimens [26]. These results are in line with those reported in the study carried out by Navarrete-Opazo et al. (2021) in the US, who observed longer hospitalisations for a population with RDs than for patients with common conditions [30].

Our results showed a decreasing annual trend of hospitalisation in the patients with RDs from 2009 to 2018, in terms of both hospitalisation rates and in inpatient proportions. The decrease in the overall hospitalisation rate appears to have been driven by the decrease in the day-case hospitalisation rate observed from 2011 (323.9) to 2018 (198.7), while the ordinary hospitalisation rate showed a steady trend from 2014. These results might suggest the better planning of day-case admissions, greater use of outpatient healthcare, or improvements in treatment regimens. 

Over the past decade, advances in diagnostic testing leading to early diagnosis may have prevented the need for unnecessary and inappropriate hospital admissions [31,32]. Moreover, the significant research progress in treating genetic diseases, which account for 80% of total RDs, and the development and approval of new orphan drugs, may have modified specific care pathways and reduced the disease burden on patients, with a prolongation of their survival and, above all, an improvement in their quality of life [33,34,35].

For the average LOS, we observed a difference of almost one day across the 2009–2018 period, which has major implications for the healthcare costs for RD patients, especially if considered over a long period of time. The results for the hospitalisation rate and average LOS could be a starting point for studies on the costs of hospitalisation. Studies on the burden of RDs and on their costs are important to plan the distribution of resources, which are usually allocated using a prevalence-based criterion, thus disadvantaging the allocation of funding for the prevention, diagnosis, and treatment of RDs.

Concerning the analysis by RD group, we observed the highest overall hospitalisation rate for the patients with rare metabolic diseases (equal to 903.3 per 1000 cases), which was affected by the fact that the highest day-case hospitalisation rate was also observed for these patients (619.0 per 1000 cases, which is at least 2–3-fold higher than the other groups of RDs). These results may be attributable to patients suffering from inherited metabolic diseases, for which they received multidisciplinary medical hospital service for diet-, drug-, and enzyme-replacement therapy, and neurodevelopmental and psychological evaluations [36]. 

The proportions of inpatients by group of RDs, observed in our study, are in line with those reported by Valent et al. (2019) [21], who carried out a study based on the rare-disease registry of Friuli Venezia Giulia, a region in Northern Italy monitoring the same groups of rare diseases, as reported by the Italian Ministry of Health. The reproducibility of the results in two different Italian regions, based on data of the same nature, confirms the importance of the applied methodological approach, allowing the integration of population-based registry data with administrative healthcare data, and its potential for health planning and research.

Emergency admissions were found to be a major burden for patients affected by metabolic diseases (63.4%). This result could be attributable to patients suffering from hereditary hemochromatosis and amyloidosis, which are the most frequent (20.1% and 13.6%, respectively) diseases in this group, and for the first ones emergency hospitalisations are often related to complications caused by the visceral overload of iron, such as cirrhosis, or to episodes of acute liver failure. For the patients with amyloidosis, the emergency admissions could have been due to multiple systemic complications involving the brain, kidney, and heart [37,38,39]. 

Among the patients suffering from rare respiratory diseases and rare digestive disorders, we found the highest percentage of ordinary admissions. This finding might be explained by the composition of the respiratory- and digestive-diseases groups involved in our study and, in particular, by the nature of the most frequent diseases in each group. In the group of respiratory diseases, sarcoidosis was the most frequent, since 45.4% of cases in this group were affected by this rare disease. Sarcoidosis is a chronic disease that mainly affects the lungs, although granulomatous infiltration often also involves the heart, eyes, and nervous system, resulting in significant morbidity and mortality [40,41]. For the group of digestive disorders, achalasia, the most frequent disease (64.4%), is characterised by dysphagia associated with regurgitation of undigested solid and liquid foods; therefore, patients are often admitted to hospital for endoscopy or surgical treatments [42].

In general, the results presented here could stimulate studies on specific rare diseases, which would allow more detailed investigations on hospitalisation, both to assess the healthcare burden of disease and to evaluate eventual associations with survival and therapeutic regimens [43].

The major strength of this epidemiological study is that it made it possible to examine the inpatients’ characteristics and the trends in the hospitalisation of the patients with RDs by using population-based registry data linked to their hospital discharge data. Using this multi-database approach, we provided a reliable estimation of the healthcare burden in a large cohort of patients diagnosed with a RD in a defined geographical area. Moreover, the results of this population-based cohort study showed that patients with RDs had a higher number of hospital admissions and a longer average LOS than the general population. Therefore, these data provide evidence of the increased costs of hospitalisation for RDs and their economic impact on healthcare. 

The limitation of the study is that the diseases monitored in the RDRT are those monitored by the Italian law (Decree of the President of the Council of Ministers, 01/2017) and endowed with a code of exemption from co-payment (see Appendix A). For this reason, although this study reports the healthcare burden referring to a group of more than 500 RDs, it cannot be considered a comprehensive assessment of all known RDs, as reported by Orphanet. Another limitation is that a reliable comparison between our results and those of other international studies is not easy because of the different groups of RDs considered, given that our study was based on the list of diseases monitored by the Italian law.

## 5. Conclusions

These findings add important knowledge on the hospitalisation profile of RD patients in terms of hospitalisation rates, the main discharge diagnoses, the proportion of inpatients, the types of admission, and the average length of stay. These indicators were produced by applying a multi-database approach using a population-based registry in synergy with a regional hospital discharge database. This method could be considered a useful approach to assessing the burden of RDs in a specific geographical area, relying on a significant cohort of cases. The method produced results that can support healthcare decision making and health-policy planning in order to improve the quality of care for RD patients and to optimise the allocation of health-care resources to reduce the burden of RDs.

## Figures and Tables

**Figure 1 ijerph-19-07553-f001:**
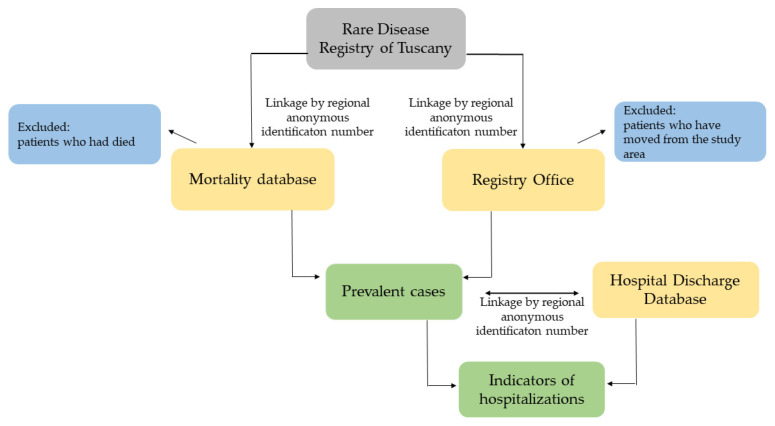
Flowchart of the selection process.

**Figure 2 ijerph-19-07553-f002:**
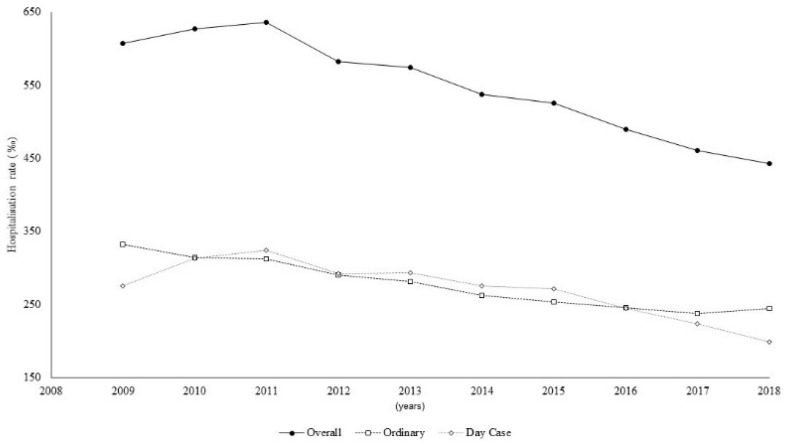
Hospitalisation time trend, overall and by type of admission.

**Figure 3 ijerph-19-07553-f003:**
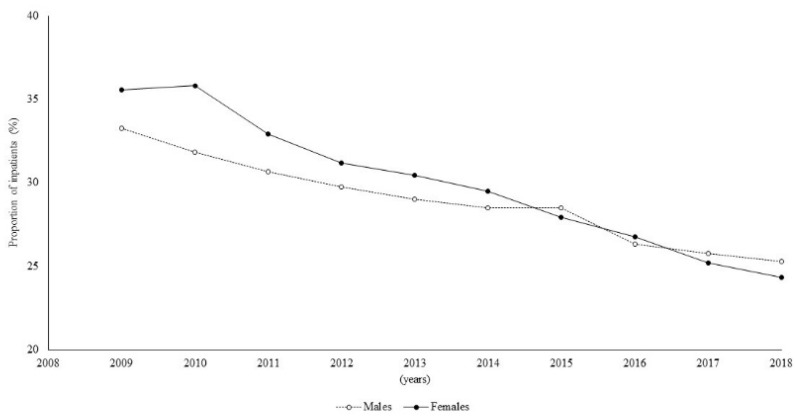
Time trend of proportion of inpatients by sex.

**Figure 4 ijerph-19-07553-f004:**
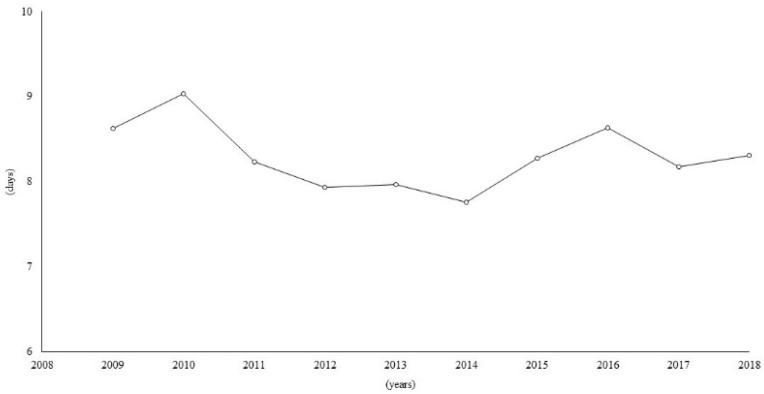
Time trend of average length of stay (expressed in days) in hospital.

**Figure 5 ijerph-19-07553-f005:**
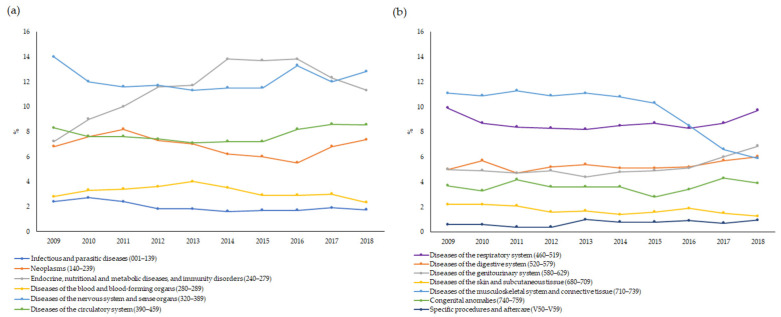
(**a**,**b**) Time trend of main causes of hospitalisation by ICD-9-CM group.

**Table 1 ijerph-19-07553-t001:** Main discharge diagnosis for overall inpatients in 2018.

Main Discharge Diagnosis	ICD-9-CM	Overall Hospitalisations *n* (%)	Ordinary Admissions *n* (%)	Day-Case Admissions *n* (%)
Infectious and parasitic diseases	001–139	138 (1.7)	110 (2.5)	28 (0.8)
Neoplasms	140–239	586 (7.4)	352 (8.1)	234 (6.5)
Endocrine, nutritional and metabolic diseases, and immunity disorders	240–279	899 (11.3)	152 (3.5)	747 (20.8)
Diseases of the blood and blood-forming organs	280–289	185 (2.3)	49 (1.1)	136 (3.8)
Diseases of the nervous system and sense organs	320–389	1021 (12.8)	416 (9.5)	605 (16.8)
Diseases of the circulatory system	390–459	681 (8.6)	600 (13.7)	81 (2.3)
Diseases of the respiratory system	460–519	774 (9.7)	718 (16.4)	56 (1.6)
Diseases of the digestive system	520–579	480 (6.0)	339 (7.8)	141 (3.9)
Diseases of the genitourinary system	580–629	547 (6.9)	266 (6.1)	281 (7.8)
Diseases of the skin and subcutaneous tissue	680–709	100 (1.3)	51 (1.2)	49 (1.4)
Diseases of the musculoskeletal system and connective tissue	710–739	469 (5.9)	270 (6.2)	199 (5.5)
Congenital anomalies	740–759	311 (3.9)	138 (3.2)	173 (4.8)
Specific procedures and aftercare	V50–V59	76 (1.0)	34 (0.8)	42 (1.2)
Other *		1694 (21.3)	871 (19.9)	823 (22.9)

Abbreviation: ICD-9-CM, International Classification of Diseases, 9th Revision, Clinical Modification. * Other main discharge diagnoses refer to: Mental disorders (290–319), complications of pregnancy, childbirth, and the puerperium (630–679), certain conditions originating in the perinatal period (760–779), symptoms, signs, and ill-defined conditions (780–799), injury and poisoning (800–899), supplementary classification of factors influencing health status and contact with health services (V01–V89, except V50–V59), supplementary classification of external causes of injury and poisoning (E800–E999).

**Table 2 ijerph-19-07553-t002:** Number of cases, proportions of inpatients, admission type (ordinary/day-case), emergency admission, average LOS and DRG type (medical/surgical), by group of rare diseases.

Groups of Rare Diseases	Cases *n* (%)	Inpatients *n* (%)	Admission Type *n* (%)		Emergency Admission * *n* (%)	Average LOS (Days)	DRG Type *n* (%)	
			Ordinary	Day-Case			Medical	Surgical
Neoplasms	796 (4.5)	200 (25.1)	163 (44.5)	203 (55.5)	62 (38.0)	6.1	255 (69.7)	111 (30.3)
Endocrine diseases	858 (4.8)	105 (12.2)	71(45.5)	85 (54.5)	27 (38.0)	6.6	117 (75.0)	39 (25.0)
Metabolic diseases	1344 (7.6)	577 (42.9)	382 (31.5)	832 (68.5)	242 (63.4)	7.5	1079 (88.9)	135 (11.1)
Immune system disorders	1090 (6.1)	236 (21.7)	231 (53.5)	201 (46.5)	116 (50.2)	9.7	328 (75.9)	104 (24.1)
Diseases of the blood and blood-forming organs	1027 (5.8)	233 (22.7)	190 (44.2)	240 (55.8)	101 (53.2)	7.6	345 (80.2)	85 (19.8)
Peripheral and central nervous system disorders	3831 (21.5)	969 (25.3)	1060 (63.2)	616 (36.8)	605 (57.1)	9.4	1248 (74.5)	428 (25.5)
Disorders of the eye and adnexa	1161 (6.5)	100 (8.6)	86 (68.8)	39 (31.2)	41 (47.7)	5.5	60 (48.0)	65 (52.0)
Circulatory system diseases	1591 (8.9)	354 (22.3)	393 (71.5)	157 (28.5)	220 (56.0)	6.9	349 (63.5)	201 (36.5)
Respiratory diseases	979 (5.5)	274 (28.0)	403 (82.1)	88 (17.9)	223 (55.3)	8.4	357 (72.7)	134 (27.3)
Digestive disorders	190 (1.1)	42 (22.1)	58 (80.6)	14 (19.4)	33 (56.9)	9.3	46 (63.9)	26 (36.1)
Diseases of the genitourinary system	368 (2.1)	117 (31.8)	125 (52.7)	112 (47.3)	58 (46.4)	6.7	187 (78.9)	50 (21.1)
Diseases of the skin and subcutaneous tissue	983 (5.5)	184 (18.7)	177 (71.7)	70 (28.3)	101 (57.1)	9.5	152 (61.5)	95 (38.5)
Diseases of the musculoskeletal system and connective tissue	1320 (7.4)	382 (28.9)	434 (68.0)	204 (32.0)	200 (46.1)	8.7	469 (73.5)	169 (26.5)
Congenital anomalies, chromosomal aberrations and genetic syndromes	2237 (12.6)	657 (29.4)	583 (44.5)	727 (55.5)	234 (40.1)	8.1	1030 (78.6)	280 (21.4)

Abbreviations: LOS, length of stay; DRG, diagnosis-related groups. * Proportion of emergency admissions referred to the total of ordinary admissions for each rare disease group.

**Table 3 ijerph-19-07553-t003:** Hospitalisation rate (per 1000 cases), overall, by type of admission, and by group of rare diseases.

Groups of Rare Diseases	Hospitalisation Rate (95% CI)		
	Overall	Ordinary Admissions	Day-Case Admissions
Neoplasms	459.8 (413.9–509.4)	204.8 (174.5–238.7)	255.0 (221.1–292.6)
Endocrine diseases	181.8 (154.4–212.7)	82.7 (64.6–104,4)	99.1 (79.1–122.5)
Metabolic diseases	903.3 (853.2–955.5)	284.2 (256.4–314.2)	619.0 (57.7–662.6)
Immune system disorders	396.3 (359.8–435.5)	211.9 (185.5–241.0)	184.4 (159.8–211.7)
Diseases of the blood and blood-forming organs	418.7 (380.0–437.6)	185.0 (159.6–213,3)	233.7 (205.1–265.2)
Peripheral and central nervous system disorders	437.5 (416.8–458.9)	276.7 (260.3–293.9)	160.8 (148.3–174.0)
Disorders of the eye and adnexa	107.7 (89.6–128.3)	74.1 (59.2–91.5)	33.6 (23.9–45.9)
Circulatory system diseases	345.7 (317.4–357.8)	247.0 (223.2–272.7)	98.7 (83.8–115.4)
Respiratory diseases	501.5 (458.1–547.9)	411.6 (372.4–453.9)	89.9 (72.1–110.7)
Digestive disorders	378.9 (296.5–477.2)	305.2 (231.8–394.6)	73.7 (40.3–123.6)
Diseases of the genitourinary system	644.0 (564.6–731.4)	339.7 (282.7–404.7)	304.3 (250.6–366.2)
Diseases of the skin and subcutaneous tissue	251.3 (220.9–284.6)	180.1 (154.4–208.6)	71.2 (55.5–90.0)
Diseases of the musculoskeletal system and connective tissue	483.3 (446.5–522.3)	328.8 (298.6–361.2)	154.5 (134.1–177.3)
Congenital anomalies, chromosomal aberrations and genetic syndromes	585.6 (554.3–618.2)	260.6 (239.9–282.6)	324.9 (301.8–349.5)

## Data Availability

The data that support the findings of this study are available from Regione Toscana, but restrictions apply to the availability of these data, which were used under license for the current study, and are not publicly available. However, the data are available from the authors upon reasonable request and with the permission of Tuscany Region.

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
