# Peer review of "Healthcare Burden of Rare Diseases: A Population-Based Study in Tuscany (Italy)"

_ijerph, 2022, doi:10.3390/ijerph19137553_

Round 1

Reviewer 1 Report

This is a highly relevant topic, the data base is sound and broad, the methodolgoy is appropriate and the findings are relevant. I recommend publishing this paper. I have only minor issues to consider:

  • use of "personalized care": The term "personalized" care or medicine has a specific meaning going beyond what you mean. Might it be better to use "tailored"?
  • resources: I wonder why you do not invest more writing about the economic consequences of your findings. Your facts clearly call for more funding of prevention, diagnoses and therapy of rare diseases. At given funds, this will even call for less funding of more common diseases. The allocation question must be stressed more.
  • The quality of Figures seems poor.
  • Why do you use ICD-9 for data from 2018? I assume you had ICD-10 at that time?
  • minor typos (e.g. page 7: "280 (21.4)" instead of "280 21.4)"
  • consider your conclusions. Based on such a strong evidence you can be more brave than that.

Author Response

Reviewer #1

This is a highly relevant topic, the data base is sound and broad, the methodology is appropriate and the findings are relevant. I recommend publishing this paper. I have only minor issues to consider:

use of "personalized care": The term "personalized" care or medicine has a specific meaning going beyond what you mean. Might it be better to use "tailored"?

Answer

According to the reviewer suggestion “personalized” has been changed into “tailored” in the revised manuscript (page 1 lines 12, 46; page 11 line 275)

resources: I wonder why you do not invest more writing about the economic consequences of your findings. Your facts clearly call for more funding of prevention, diagnoses and therapy of rare diseases. At given funds, this will even call for less funding of more common diseases. The allocation question must be stressed more.

Answer

We completely agree with the Reviewer. Unfortunately, data about costs of hospitalization were not available for this study. However, following the reviewer suggestion, we added a short paragraph in Discussion about the importance of funding allocated to rare diseases following criteria that are not only based on the disease prevalence (page 11, lines 304-309).

The quality of Figures seems poor.

Answer

We agree, Figures attached in the text seem to have a poor quality. However, we uploaded all the figures at 300dpi resolution during the submission process. We wait for an eventual comment by the Editor and we are ready to produce Figures at a higher quality.

Why do you use ICD-9 for data from 2018? I assume you had ICD-10 at that time?

Answer

Italian hospital discharge databases use ICD-9-CM instead of ICD-10 as reported in Methods (lines 100-102) [Reference: Ministero della Sanita Decreto 27 ottobre 2000 , n. 380 Regolamento recante norme concernenti l'aggiornamento della disciplina del flusso informativo sui dimessi dagli istituti di ricovero pubblici e privati].

minor typos (e.g. page 7: "280 (21.4)" instead of "280 21.4)"

Answer

it has been corrected in this revised version (page 8, Table 2.).

consider your conclusions. Based on such a strong evidence you can be more brave than that.

Answer

As suggested by the Reviewer we have expanded the text in the Conclusions section as follow: “…results that can support healthcare decision making and health policy planning in order to improve the quality of care for RD patient and to optimize the allocation of health-care resources to reduce the burden of RDs (page 14, from line 382).

Reviewer 2 Report

The authors summarized the prevalence of RD in Tuscany (Italian region). The manuscript contains clear tables of descriptive statistics about RD. However, there are some concerns. 

  1. Title: The title seems to be too broad. Because this study uses a local database in Italia, the title must contain the region's information.
  2. In the retrospective cohort study, the selection process of point prevalence data from the database is commonly reported (as the flowchart) for reproducible analysis. The flowchart of the selection process is necessary. 
  3. The time trend of hospitalization and its interpretation is interesting. If possible, the international comparison (with the other countries) will enrich the results. More detailed discussion and investigations are necessary in Introduction or Discussion section. 
  4. For analyzing the distribution of the main discharge diagnosis, the authors only considered the data in 2018. Is there any special reason? The distributional changes for ten years and longitudinal data analysis would be also interesting. 
  5. In this manuscript, the prevalence of RDs is the main interest. Some discussion about the time trend of incidence of RD would be very helpful for readers. 

Author Response

Reviewer #2

The authors summarized the prevalence of RD in Tuscany (Italian region). The manuscript contains clear tables of descriptive statistics about RD. However, there are some concerns.

Title: The title seems to be too broad. Because this study uses a local database in Italia, the title must contain the region's information.

Answer

The title has been changed accordingly to the Reviewer’s comment “Healthcare burden of rare diseases: a population-based study in Tuscany (Italy)”.

In the retrospective cohort study, the selection process of point prevalence data from the database is commonly reported (as the flowchart) for reproducible analysis. The flowchart of the selection process is necessary.

Answer

We agree with Reviewer, so we are now providing a flowchart as Figure 1 and for this reason, all the Figures have been renumbered in the revised manuscript (page 3, line 92; page 5 lines 172,180).

The time trend of hospitalization and its interpretation is interesting. If possible, the international comparison (with the other countries) will enrich the results. More detailed discussion and investigations are necessary in Introduction or Discussion section.

Answer

We agree with the Reviewer comment, but we found difficult making reliable comparison with other publication in literature since the group of RDs we considered is different from that of other studies we cited in the text. The only reliable comparison is that reported in the Discussion, because it is based on another Italian Registry considering the same RDs, being it based on the list monitored by the Italian law. For example, in their interesting paper, Walker et al (Western Australia) found an average LOS of 5.5 days in 2010 that is lower than ours (9 days) in the same year. This discrepancy is probably explained with the different group of RD considered in the study. A similar discrepancy was observed with Navarrete-Opazo et al. (USA, 2021) who found in 2016 an average LOS of 6.3 days for overall RD patients, whereas in the same year we found an average LOS of 8.6 days for RD patients. For this reason, among the limitations of the study it has been reported that a reliable comparison between our results and other international studies is not easy, because of the different groups of RDs considered (page 13, lines 369-372).

For analyzing the distribution of the main discharge diagnosis, the authors only considered the data in 2018. Is there any special reason? The distributional changes for ten years and longitudinal data analysis would be also interesting.

Answer

We agree with the Reviewer, so the time trend of main discharge diagnosis has been added in the results section in the  revised version of the manuscript, reporting the results in Figure 5 a-b (page 5 lines 181-190; page 6, line 191).

In general, we chose to make a focus on groups of disease limiting the analysis to the last year available (2018). Otherwise, it would have been complicated to discuss all the results by group and by year.

In this manuscript, the prevalence of RDs is the main interest. Some discussion about the time trend of incidence of RD would be very helpful for readers.

Answer

The main interest of this study was to evaluate the hospitalization of RD patients calculating indicators for the overall RDs (10 year-trends 2009-2018) and by groups of RD (only 2018). However, the Reviewer’s comment is very interesting and we looked at the time trend of incidence in order to understand if an eventual variation in incidence across the years of the wider groups of RDs could have driven the decreasing trend of hospitalization. Such analysis showed that the incidence remains almost stable across the years for all the RDs with a slight increase only for some of them, which cannot explain the decreasing trend in day-case hospitalization observed for the overall RDs. For this reason, we can confirm our hypothesis that the decreasing trend of day-case hospitalization might be due to the better planning of day-case admissions, the greater use of outpatient healthcare, the improvement in the treatment regimen or to a combination of them.

Reviewer 3 Report

This paper appears to be in the form of a report on the medical use of patients with rare diseases. Apart from the rarity of the data, there are few points that may be of particular interest to international readers.

The points that need improvement are described below.

-       The core of this paper is thought to be the analysis data. The data used were not designed for epidemiological investigation, but a combination of administrative data and hospital data. Therefore, data cleaning will be an important task. You need to describe for this

-       It will be easy for readers to understand if there is a data organization flow chart to help understand the process of combining the data and deriving the final analysis data.

-       It is difficult to identify a specific disease because it is classified only by the DRG code. In order to develop interventions based on the various indicators used in this study, an approach to specific diagnostic names is required. Was there no such possibility in this study?

-       Why are detailed indicators of hospitalization limited to the 2018 cohort?   -       Because not all readers understand the Italian health care system, it seems that a description of the health care system including Italy's reimbursement system is necessary.   -       The sentence on lines 327 – 330, which is part of the research findings confirmed by the authors is sufficiently predictable. Isn't the lack of a more detailed discussion due to the use of the Hospital Discharge Database rather than the hospital treatment data?  

Moreover, the results of this population-based cohort study showed that patients with RDs had a higher number of hospital admissions and a longer average LOS than the general population. “

Author Response

Reviewer #3

This paper appears to be in the form of a report on the medical use of patients with rare diseases. Apart from the rarity of the data, there are few points that may be of particular interest to international readers.

The points that need improvement are described below.

-       The core of this paper is thought to be the analysis data. The data used were not designed for epidemiological investigation, but a combination of administrative data and hospital data. Therefore, data cleaning will be an important task. You need to describe for this

Answer

Our study is based on a combination of data from: the population-based registry of rare disease, the mortality database, the Registry office database and the hospital discharge database. This kind of data are commonly used both for epidemiological investigations (see for example refs 20, 21, 24, 41, 43) and for epidemiological reports (see for example ref 29). However, in order to better explain the combination of such data we have added a flowchart as Figure 1. referred to details reported in Materials and Methods, as suggested by the Reviewer in the following point.

-      It will be easy for readers to understand if there is a data organization flow chart to help understand the process of combining the data and deriving the final analysis data.

Answer

We agree with the Reviewer. Moreover, also the Reviewer #2 suggested to produce a flowchart to better understand the process of combining data. So, we are now providing the flowchart as Figure 1. For this reason, all the Figures have been renumbered in the revised manuscript (page 3, line 92; page 5 lines 172,180).

-       It is difficult to identify a specific disease because it is classified only by the DRG code. In order to develop interventions based on the various indicators used in this study, an approach to specific diagnostic names is required. Was there no such possibility in this study?

Answer

The diseases involved in our study are reported in Supplementary Material and grouped according to 16 nosological groups. Results by disease groups are also reported and commented in Discussion (lines 288-326). For what concern DRG, DRG types were only used to report the proportion of RD patient undergoing (DRG surgical) or not-undergoing (DRG medical) to a surgical procedure (Table 2).

-       Why are detailed indicators of hospitalization limited to the 2018 cohort?  

In general, we chose to make the focus on groups of disease limiting the analysis to the last year available (2018). Otherwise it would have been complicated to discuss all the results by group and by year. However, the time trend of main discharge diagnosis has been added in the Results section to this revised version of the manuscript, as also requested by another Reviewer (page 5 lines 181-190; page 6, line 191).

-       Because not all readers understand the Italian health care system, it seems that a description of the health care system including Italy's reimbursement system is necessary.  

Answer

A short description of the Italian reimbursement system in relation to Diagnosis Related Group has been added in Materials and Methods as follow: “Each hospital discharge record according to inpatient’s age gender, main diagnosis, secondary diagnoses, surgical or not surgical procedures performed, and status of the patients at discharged, is assigned to a specific Diagnosis Related Group (DRG) [25]. In 1994, the Ministry of Health adopted the DRG classification for the inpatients Reimbursement of the NIHS (D.M. 15 Aprile 1994). In this study we used DRG information to identify patients undergoing to a surgical procedure (DRG surgical) and those in which a surgical treatment was not performed (DRG medical)” (page 3, lines 109-115).

-     The sentence on lines 327 – 330, which is part of the research findings confirmed by the authors is sufficiently predictable. Isn't the lack of a more detailed discussion due to the use of the Hospital Discharge Database rather than the hospital treatment data?  “Moreover, the results of this population-based cohort study showed that patients with RDs had a higher number of hospital admissions and a longer average LOS than the general population. “

Answer

We think that the sentence on lines 327-330 can be interesting as a public health message, we know that is a predictable result and for this reason we already added in the original version of the manuscript that such result is somehow expected (lines 274-275). However, this result confirms the ones observed in a US study on the impact of health-care utilization by rare disease patients. Another Reviewer asked us to add comments about the burden of RDs and its relation to the costs and to funding allocation. We think that these additional comments enrich the discussion and can help in better understanding the observed results.

Round 2

Reviewer 3 Report

As the authors pointed out at the limitations of the study, the DRG is not a code for classifying diseases, but a classification for determining the level of reimbursement. There are fundamental limitations in analyzing medical use using DRG. Even if they belong to the same DGR group, different responses and appropriate policies should be presented according to the type or severity of the disease. The paper is to group the diseases at the average level of resource use and to describe the average hospital administration data for those disease groups. This point should be clearly stated.

Author Response

Reviewer #3 R2:

As the authors pointed out at the limitations of the study, the DRG is not a code for classifying diseases, but a classification for determining the level of reimbursement. There are fundamental limitations in analyzing medical use using DRG. Even if they belong to the same DGR group, different responses and appropriate policies should be presented according to the type or severity of the disease. The paper is to group the diseases at the average level of resource use and to describe the average hospital administration data for those disease groups. This point should be clearly stated.

According to the reviewer’s comment, we reworded, in Methods section, the paragraph at lines 104-111, clearly stating that we used the DRG types (surgical/medical) only to classify the type of discharge.